# Thermal Stabilisation of Lysozyme through Ensilication

**DOI:** 10.3390/molecules29174207

**Published:** 2024-09-05

**Authors:** Reveng A. Abdulkareem, Aswin Doekhie, Nikoletta Fotaki, Francoise Koumanov, Charlotte A. Dodson, Asel Sartbaeva

**Affiliations:** 1Department of Life Sciences, University of Bath, Claverton Down, Bath BA2 7AY, UK; 2Department of Medical Education, College of Medicine, University of Duhok, 1006 AJ Duhok, Kurdistan Region, Iraq; 3Department of Chemistry, University of Bath, Claverton Down, Bath BA2 7AY, UK; 4Department for Health, Centre for Nutrition, Exercise and Metabolism, University of Bath, Claverton Down, Bath BA2 7AY, UK; 5Ensilicated Technologies Ltd., Science Creates St. Philips, Albert Road, St. Philips, Bristol BS2 0XJ, UK

**Keywords:** lysozyme, ensilication, protein stabilisation, sol–gel technology

## Abstract

Protein therapeutics, vaccines, and other commercial products are often sensitive to environmental factors, such as temperature and long-term storage. In many cases, long-term protein stability is achieved by refrigeration or freezing. One alternative is the encapsulation of the protein cargo within an inert silica matrix (ensilication) and storage or transport at room temperature as a dry powder. In this paper, we test the effect of three commonly used biological buffers on the ensilication, storage, and desilication of the enzyme lysozyme. We show that ensilication protects lysozyme from heat (100 °C for 1 h) and during storage (18 months at room temperature). The choice of ensilication buffer has little effect on the activity of lysozyme after desilication. Our results provide confidence in the continued pursuit of ensilication as a methodology for protein stabilisation and in its compatibility with biological buffers.

## 1. Introduction

The sensitivity of many proteins, including protein therapeutics and vaccines, to thermal stress, desiccation, and lyophilisation has led to the use of cold storage for proteins in the healthcare sector. Maintenance of this temperature regime from manufacture to patient administration is termed the cold chain and can be difficult to achieve in regions with no or interrupted power supplies [1,2,3,4,5]. Even in countries where the cold chain can routinely be achieved there is a considerable wastage of materials [5], and the refrigeration or freezing of a protein from manufacture to use is more expensive and more energy-intensive than storage at room temperature.

One strategy to address these problems is to encase the protein molecules in a silica shell using modified solution–gelation (sol–gel) technology (ensilication) [6,7,8]. Ensilication is distinct from technologies in which proteins are adsorbed onto inorganic porous materials (including silica) [6] due to the expected pore size (which–in the case of ensilication–is smaller than the size of the cargo protein) and is also distinct from biomineralisation in either calcium phosphate [9] or manganese phosphate [10] nanoparticles. Coating and other nanoparticle technologies complement more traditional approaches to stabilise biotherapeutics such as formulation (e.g., optimising buffer pH, presence of excipients), PEGylation, and protein engineering of the therapeutic itself [11,12]. While these latter strategies often prolong the shelf life of a therapeutic, they rarely remove the requirement for a cold chain.

Ensilication is based on the acid-catalysed hydrolysis of tetraethyl orthosilicate (TEOS) to form silicic acid, Si(OH)_4_, which then undergoes a condensation reaction, polymerising to form a solid silica matrix. If hydrolysed silica is mixed with protein solution, the silica will polymerise around protein molecules to create a solid mesh, effectively encapsulating the protein in a reaction that proceeds at ambient temperatures (or lower). This can subsequently be filtered, resulting in a dry powder of encapsulated protein. Cargo protein can be released from encapsulation (desilicated) by hydrolysis of the silica in the presence of fluoride.

The reversible nature of the ensilication–desilication process and the formation of the silica matrix around the protein molecules are strengths of this approach [6]. Recent work has shown that both ensilicated model proteins and the *tuberculosis* antigen 75b retain their native fold upon desilication, even if they have been subject to heat or ageing while ensilicated [13,14]. Most recently, an ensilicated tetanus toxin C fragment, which had been heat-treated at 80 °C for 2 h, was shown to provoke a specific immunogenic response in mice upon desilication [15]. The inert nature of silica is also expected to provide chemical protection to the cargo protein.

Our previous work focused on the protective effects of a silica coating on protein stability but did not detail the effect of buffer salts and other electrolytes on the ensilication process itself, the morphology of the ensuing nanoparticles, or the preservation of the ensilicated cargo. It is known that particle size and aggregation behaviour (degree of network formation) during silica polymerisation and gelation depend on multiple factors, including pH, temperature, concentration, and the identity of buffer salts [8,16]. Silica polymerisation may also be influenced by electrostatic interactions between the surface of the cargo protein and the silica nanoparticles and monomers. 

In this work, we set out to determine the effect of three different protein buffers (50 mM Tris-HCl (pH 7.2), 50 mM sodium phosphate (pH 7.2), and phosphate buffer saline (PBS)) on the ensilication of the model protein lysozyme. These buffers differ in a number of ways, including the buffer species, total ionic strength, and the presence or absence of chloride ions. We compare the effect of ensilication buffer on the morphology of the particles, on the efficiency of protein encapsulation and desilication, and on the protection from thermal stress and ageing given to the protein cargo.

Our results show that efficient protein encapsulation occurs under all conditions tested. We achieved smaller particles of a more uniform diameter in the absence of chloride ions, and these particles had a slightly higher ratio of protein/silica when compared with the other buffer conditions. Protein retrieval after desilication and maintenance of lysozyme catalytic activity were unaffected either by ensilication buffer conditions or by stresses experienced by the ensilicated protein.

## 2. Results and Discussion

### 2.1. Effect of Buffer on the Ensilication Process

Lysozyme powder was dissolved in buffer, ensilicated, and the presence of protein in each solid powder was confirmed using FTIR (Figure 1). All ensilicated samples exhibited characteristic protein peaks corresponding to amide I (C=O stretching) and amide II (C-N stretching and N-H bending) vibrations. Peaks at wavenumbers 1054 cm^−1^ and 942 cm^−1^, indicative of Si-O-Si and Si-OH stretching, were also evident, indicating the successful encapsulation of lysozyme within the silica material. 

A higher spectral energy was observed for the amide I and II bands in the ensilicated samples (peak positions 1654 cm^−1^ and 1540 cm^−1^) compared with untreated lyophilised lysozyme powder (1643 cm^−1^ and 1515 cm^−1^). This could be due to a number of factors, including differences in hydrogen bonding, hydration (all samples are nominally dry powders), or simply the different electronic environment of surface amides in the presence of a silica shell [17,18].

### 2.2. Morphology of Silica Nanoparticles

In order to determine the effect of ensilication buffer on powder morphology, we carried out field emission scanning electron microscopy (FE-SEM) of our ensilicated samples. Silica polymerisation in the absence of both protein and buffer (Figure 2A) or in the absence of protein but presence of buffer (Figure 2B) did not result in discrete nanoparticles. For ensilication (in the presence of both buffer and protein; Figure 2C–E), reactions in PBS and Tris-HCl resulted in particles with both a larger average diameter and an overall broader distribution of particle diameters compared with phosphate buffer alone (Figure 2F and Table 1). Differences in particle size cannot be attributed to pH, nonspecific shielding by either positively or negatively charged buffer species, or overall ionic strength (Table 1). Instead, we tentatively assign these differences to a specific effect of the chloride ion, which is known to decrease the solubility of amorphous silica [19], potentially leading to the rapid formation of aggregated silica nanoparticles [8].

### 2.3. Effect of Buffer on Ensilication Efficiency and Drug Loading

In order to determine the relative ratios of protein and silica in our samples, we calculated the percentage drug loading for each of our conditions (Figure 3A). Drug loading is defined as the fractional amount of drug in a drug/carrier mixture and, in our case, determines the proportion of protein in our ensilicated samples by mass.

The highest drug loading (~47%) was observed in phosphate buffer, while the lowest (~30%) was observed in PBS. This is consistent with the growth of silica nanoparticles by monomer addition (around protein molecules) at low salt concentrations compared with particle aggregation (of potentially small silica-only particles) at higher [NaCl] [8].

The encapsulation efficiency—in our case also equal to the yield of the ensilication and desilication processes—was found to be consistently high across all buffer conditions. Around 80% of all lysozyme in the initial samples was retrieved in soluble form after desilication (Figure 3B).

We also determined the effect of thermal stress and thermal stress after milling to a fine powder on both drug loading and ensilication efficiency. While we would not expect treatment while ensilicated in powder form to affect the protein/silica ratio in the ensilicated sample per se, this ratio is determined after desilication, and so any protein degradation or formation of insoluble aggregates while ensilicated would result in an apparent decrease in this parameter compared with control. Additionally, any destruction of the encapsulating silica shell would result in an apparent increase in drug loading compared with untreated samples.

As expected, there was a negligible change in drug loading for samples subjected to heat treatment alone or heat treatment plus milling compared with samples stored at room temperature. There was also a negligible change in the encapsulation efficiency (overall yield of protein following desilication). 

Milling, which increases the surface area of the ensilicated powder, did not appear to impact either apparent loading or overall yield. In some ways this is logical since ensilication is expected to occur on the molecular level and the length scales of lysozyme molecules (4 nm), visible features in ensilicated material (100s of nm; Figure 2), and milled particle size (<180 μm) span several orders of magnitude. We were particularly pleased to learn that there was no change in encapsulation efficiency (overall yield of protein from encapsulation and desilication cycle), since this indicates that protein desilication is not currently limited by the accessible surface area of ensilicated material on this scale. 

We also determined the stability of silica shells for protein encapsulation over an 18-month storage period at room temperature (Figure 3). No changes were observed in either drug loading or ensilication efficiency, indicating that the silica shells maintained their structural integrity throughout the storage period. 

### 2.4. Circular Dichroism (CD) of Treated Lysozyme

We next tested whether lysozyme ensilication and desilication result in irreversible changes to the secondary structure of the enzyme itself. The CD spectrum of untreated lysozyme (black dashed line, Figure 4) displayed characteristic α-helix peaks below 195 nm and above 205 nm, consistent with the spectrum predicted by PDB2CD [20]. The spectra of desilicated lysozyme (with and without heat treatment while ensilicated) showed only minor changes compared with the untreated sample, indicating that treatment caused no irreversible changes to the protein secondary structure.

### 2.5. Enzyme Activity of Treated Lysozyme

Finally, to determine the impact of ensilication on the functional activity of lysozyme, we compared the enzymatic activity of treated lysozyme with that of freshly dissolved, untreated control (Figure 5). All ensilicated samples retained activity after desilication, with little difference between ensilication buffer or treatment conditions for samples prepared in Tris-HCl and phosphate. Ensilicated lysozyme maintained its activity even after thermal stress, in contrast to the untreated control where 80% of activity was lost. This is consistent with previous results for lysozyme ensilicated in Tris-HCl alone [13]. Storage of ensilicated lysozyme at room temperature for 18 months had little effect on the activity of the desilicated enzyme.

Lysozyme ensilicated in PBS retained 88% of the activity of untreated control, higher than that observed in Tris-HCl- and phosphate buffer-ensilicated samples. This activity was similar (85%) after 18 months of storage at room temperature but was reduced to 65% upon thermal stress at 100 °C. 

## 3. Methodology

### 3.1. Chemical Materials

Lysozyme powder (chicken egg white), tetraethyl orthosilicate (TEOS) solution, trizma base powder, sodium fluoride powder, sodium phosphate dibasic hepta-hydrate powder, and sodium phosphate monobasic monohydrate were sourced from Sigma-Aldrich Company Ltd. (Gillingham, UK). Fisher Scientific (Loughborough, UK) provided 37% hydrochloric acid, anhydrous sodium carbonate, sodium hydrogen carbonate, sodium hydroxide pellets, and the EnzCheck Lysozyme Assay Kit (Invitrogen, Paisley, UK). Phosphate-buffer saline (PBS) tablets (8 mM Na_2_HPO_4_, 1.5 mM KH_2_PO_4_, 137 mM NaCl, 3 mM KCl) were obtained from Oxoid Limited (Basingstoke, UK). Ultra-pure (Milli-Q) laboratory-grade water was used throughout the study.

### 3.2. Ionic Strength Calculations

The ionic strength of all buffers was calculated using Equations (1)–(3).
(1)$$[AB]=\frac{1}{2}\left([A]+[B]+K_{d}\right)-\sqrt{{\left([A]+[B]+K_{d}\right)}^{2}-4[A][B]}$$
(2)$$K_{d}=10^{-pK_{a}}$$
where [*AB*] is the concentration of the complex *AB*, [*A*] is the concentration of the free buffer species *A*, [*B*] is the concentration of the free buffer species *B*, *K*_d_ is the dissociation constant of the buffer, and p*K_a_* is the literature value for the buffer in question.
(3)$$I=\frac{1}{2}\sum_{i}c_{i}z_{i}^{2}$$
where *I* is the total ionic strength, *c_i_* is the concentration of a charged species *i*, and *z_i_* is the charge on species *i*.

### 3.3. Ensilication of Lysozyme and Determination of Protein Concentration

Ensilicated lysozyme was prepared as previously described [13] with slight modifications. Briefly, 1 part of hydrolysed silica solution (10 mM HCl, 50% (*v*/*v*) TEOS in water) was mixed with 50 parts of lysozyme solution (1 mg/mL lysozyme in one of 50 mM Tris-HCl pH 7.2, 50 mM sodium phosphate pH 7.2, or PBS pH 7.4). The final ensilication solution was composed of 0.98 mg/mL lysozyme, 1% (*v*/*v*) TEOS, and one of 49 mM Tris-HCl pH 7.2, 49 mM sodium phosphate pH 7.2, or PBS pH 7.4.

This mixture was stirred for 20 min, filtered, and left to dry in an extractor for one day. The protein within the dried filtrate was desilicated by mixing it with desilication buffer (95 mM NaF-HCl, 25 mM Tris-HCl pH 4) in the ratio 1 mg:2 mL. Filtrate and buffer were stirred at room temperature for 1 h. Absorbance was then measured at 280 nm. The concentration of protein was determined using a Lambda 650S UV/Vis Spectrophotometer (Perkin Elmer, Beaconsfield, UK) and the Beer–Lambert law using a molar extinction coefficient (ε) of 37,470 mol^−1^cm^−1^ and a molecular weight of 14,313 g/mol.

### 3.4. Fourier Transform Infrared Spectroscopy (FTIR)

FTIR measurements were carried out using a Spectrum 100 FTIR Spectrophotometer (Perkin Elmer, Beaconsfield, UK). Air served as the background for analysing the powdered samples. Each sample was scanned eight times across a wavenumber range from 4000 cm^−1^ to 600 cm^−1^.

### 3.5. Field Emission Scanning Electron Microscopy (FE-SEM)

FE-SEM was carried out using a JSM-7900F FESEM (JEOL, Welwyn Garden City, UK). Samples were prepared by grinding, mounting, and gold coating.

### 3.6. Particle Sizes

The diameter of 100 randomly selected ensilicated nanoparticles was measured using the line tool within ImageJ [21] and calibrated using the image scale bar. The collected data were tabulated in ImageJ before statistical analysis as described below.

### 3.7. Statistical Analysis 

Quantitative data were analysed using the Shapiro–Wilk and Levene statistical tests to test for Gaussian distribution and homogeneity of variance respectively. This was followed by one-way ANOVA and a post hoc Tukey HSD test for comparison within groups. All analyses were carried out using SPSS software [22].

### 3.8. Percentage Drug Loading

Percentage drug loading (*DL*%) is traditionally calculated using Equation (4) [23].
(4)$$DL\%=\frac{\mathrm{mass}\;\mathrm{of}\;\mathrm{drug}\;\mathrm{incorporated}}{\mathrm{mass}\;\mathrm{of}\;\mathrm{drug}\;\mathrm{incorporated}+\mathrm{mass}\;\mathrm{of}\;\mathrm{carrier}}$$

In our context, the mass of protein incorporated is assumed to be the same as that desilicated (i.e., we assume 100% desilication efficiency), and drug loading is calculated using Equation (5).
(5)$$DL\%=\frac{m_{prot}}{m_{t}}=\frac{[\mathrm{protein}\;\mathrm{released}]\cdot V}{m_{t}}\cdot 100$$
where *m_prot_* is the mass of protein in the ensilicated sample, *m_t_* is the total mass of the ensilicated sample (i.e., the mass of protein + the mass of silica), and *V* is the volume of desilication medium used. Drug loading reflects the protein/silica ratio within an ensilicated sample, with a higher value indicating a thinner shell of silica (more protein per silica).

### 3.9. Encapsulation Efficiency

Encapsulation efficiency (*EE*%) quantifies the fraction of drug present in an initial sample that is encapsulated into a given drug–carrier complex (Equation (6) [23]). In our context, we again assume 100% sample desilication and determine the amount of protein originally encapsulated from the concentration of protein in solution after desilication (Equation (7)).
(6)$$EE\%=\frac{\mathrm{mass}\;\mathrm{of}\;\mathrm{protein}\;\mathrm{encapsulated}}{\mathrm{mass}\;\mathrm{of}\;\mathrm{protein}\;\mathrm{added}\;\mathrm{to}\;\mathrm{encapsulation}\;\mathrm{reaction}}$$
(7)$$EE\%=\frac{[\mathrm{protein}\;\mathrm{released}]\cdot V}{m_{total}}\cdot \frac{m_{ensilicated}}{m_{released}}\cdot 100$$
where *m_total_* is the total mass of protein added to the encapsulation reaction, *V* is the volume of desilication medium, *m_ensilicated_* is the total mass of the dry ensilicated material after the ensilication reaction, and *m_desilicated_* is the mass of ensilicated material used in the desilication reaction. The fraction *m_ensilicated_*/*m_desilicated_* is a scaling factor to account for the fact that only a small proportion of the ensilicated sample is used to determine encapsulation efficiency. In our context, the reported value of encapsulation efficiency is the same as the yield of the total ensilication–desilication process.

### 3.10. Thermal Stress

Ensilicated samples of lysozyme were exposed to thermal stress by being placed in an oven at 100 °C for 1 h. Untreated lyophilised lysozyme powder was used as a control. After stress, ensilicated samples were released as described above. Untreated lysozyme samples were dissolved in 50 mM Tris-HCl pH 7.2 and incubated at room temperature for 1 h (to simulate incubation during the release process).

### 3.11. Milling

Ensilicated lysozyme samples (20 mg) were either milled with a mortar and pestle into a fine powder (defined as passing through pharmaceutical sieve No. 85, nominal mesh aperture 180 μm) or left intact.

### 3.12. Circular Dichroism (CD)

All CD measurements were acquired using a Chirascan VX (Applied Photophysics, Leatherhead, UK). To ensure consistency in treatment duration, the lyophilised lysozyme powder (control) was dissolved in release buffer and incubated at room temperature for 1 h, mirroring the release time of the ensilicated samples. Following release, all samples were exchanged into 10 mM phosphate buffer pH 7 using a PD-10 desalting column, which separated protein from both the release buffer and silica. 

Samples for analysis were placed in a 0.1 mm cuvette (Stana Scientific, Hainault, UK), and wavelength scans were collected over the range 260–185 nm. Molar extinction coefficients were determined using Equation (8).
(8)$$\Delta \varepsilon =\frac{\mathrm{raw}\;\mathrm{signal}\cdot 0.1}{[lysozyme]\cdot pathlength\cdot 3298}$$
where raw signal is ellipticity in millidegrees, [*lysozyme*] is expressed in M, and *pathlength* is expressed in cm.

The secondary structure was compared with the parameters predicted by the PDB2CD website [20], using the deposited PDB structure 2W1X as input.

### 3.13. Lysozyme Activity Assay

Lysozyme activity was measured using the EnzChek Lysozyme fluorescence assay with a PHERAstar FS plate reader (BMG Labtech, Ortenberg, Germany), following the manufacturer’s instructions (kit E22013, Invitrogen, Paisley, UK). The assay utilises a fluorescently quenched lysozyme substrate from *Micrococcus lysodeikticus*, provided as part of the kit. Lysozyme activity releases the fluorophore, resulting in a fluorescence signal directly proportional to the concentration of active lysozyme [24]. Assays were carried out in the reaction buffer provided (100 mM sodium phosphate pH 7.5, 100 mM NaCl, + sodium azide as a preservative), and the activity was expressed as active lysozyme units per milligram of protein. 

Untreated control samples were dry lyophilised lysozyme powder, treated as described. After treatment, samples were dissolved in 50 mM tris (pH 7.2), incubated for 1 h at room temperature, and then assayed in phosphate buffer as described above.

## 4. Conclusions

We have previously established the ensilication of proteins as a promising strategy to protect protein samples against both desiccation and thermal stress [13,15,25]. In this study, we compare the ensilication of lysozyme in three common biological buffers (50 mM Tris-HCl pH 7.2, 50 mM phosphate pH 7.2, and PBS) and establish the robustness of the ensilication reaction in general. 

FTIR analysis confirmed the encapsulation of lysozyme within all ensilicated materials, and circular dichroism measurements indicated that ensilication caused no irreversible change to the protein secondary structure. Ensilication under all three buffer conditions effectively protected the encapsulated lysozyme from thermal stress (100 °C for 1 h) and from degradation during an 18-month storage period at room temperature.

A few buffer-specific differences in the morphology and drug loading of the silica–protein nanoparticles were observed. Most notably, ensilication in the absence of chloride ions (50 mM phosphate pH 7.2) resulted in smaller particles, which were more uniform in size. Particles created in the absence of chloride ions also had a higher ratio of protein to silica (higher drug loading). 

Overall, our results determine the robustness of the ensilication reaction to the buffer systems used for the biological sample and pave the way for more general application of this technique.

## Figures and Tables

**Figure 1 molecules-29-04207-f001:**
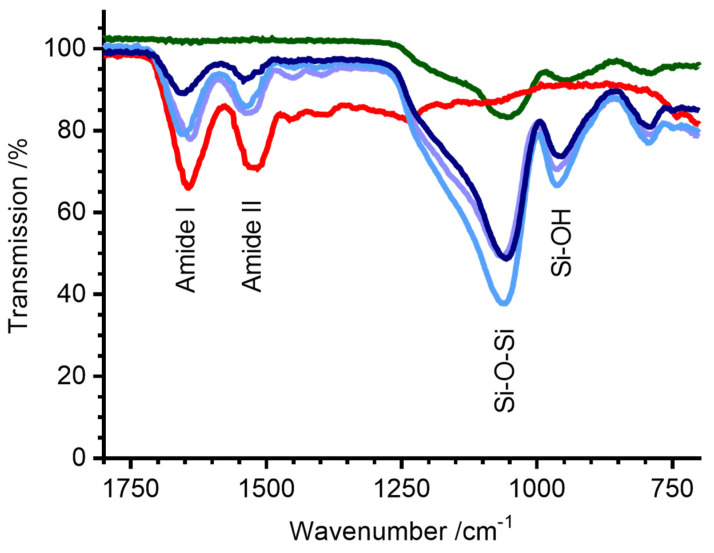
FTIR (powder) spectra of ensilicated lysozyme. Green—silica alone, red—untreated lyophilised lysozyme powder, light purple—ensilicated lysozyme (ensilication reaction in Tris-HCl buffer), light blue—ensilicated lysozyme (ensilication reaction in phosphate buffer), dark blue—ensilicated lysozyme (ensilication reaction in PBS). Amide I (C=O), Amide II (C-N), Si-O-Si, and Si-OH stretching bands are labelled.

**Figure 2 molecules-29-04207-f002:**
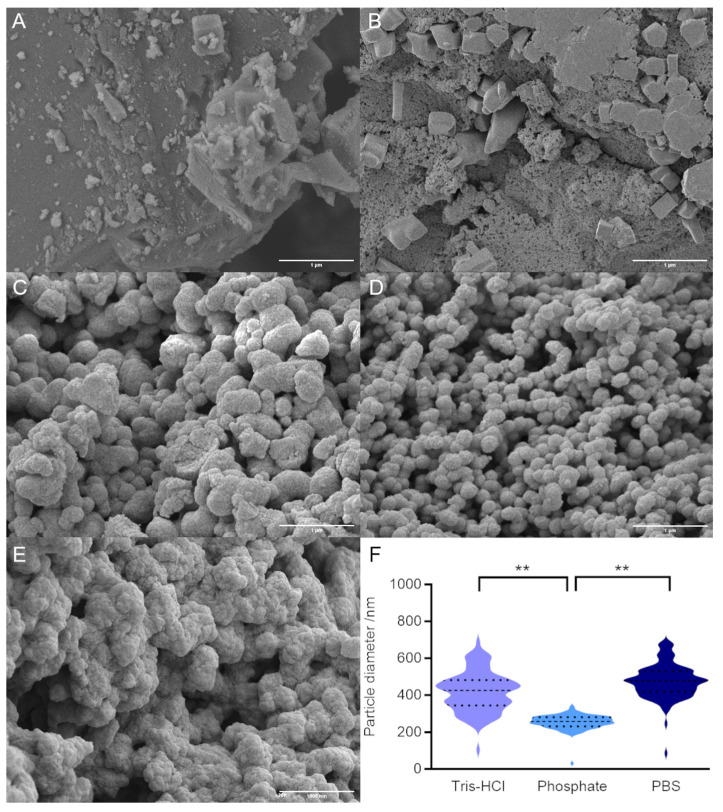
FE-SEM of ensilicated lysozyme samples. (**A**) Control reaction with silica alone (no protein, no buffer). (**B**) Control reaction with silica and PBS alone (no protein). (**C**) Lysozyme ensilicated in Tris buffer. (**D**) Lysozyme ensilicated in phosphate buffer. (**E**) Lysozyme ensilicated in PBS. Scale bar in each panel is 1 μm. (**F**) Quantification of particle diameters for ensilicated protein (100 particles). Dashed line shows mean and dotted lines show upper and lower quartiles. ** indicates significant difference (*p* < 0.01) in mean values, as determined by Tukey test.

**Figure 3 molecules-29-04207-f003:**
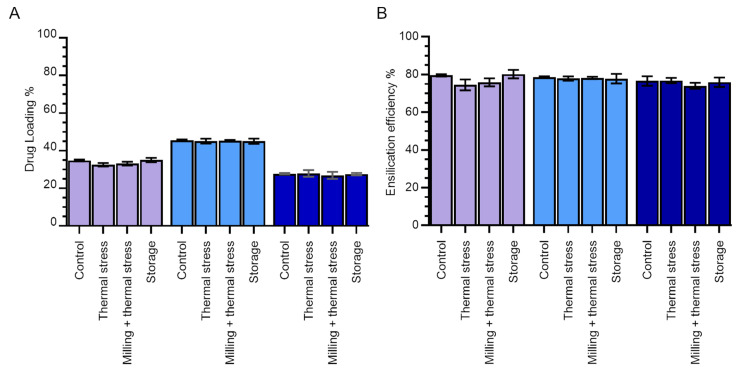
Effect of ensilication buffer and stress when ensilicated on lysozyme–silica nanoparticles. (**A**) Percentage drug loading. (**B**) Encapsulation efficiency. The effect of different stressors (long-term storage (18 months) at room temperature, thermal stress (1 h in oven at 100 °C), or milling and thermal stress combined) on the ensilicated material was compared with the control (ensilicated lysozyme dried overnight at room temperature). Purple—Tris-HCl; light blue—phosphate; dark blue—PBS. Error bars indicate the standard deviation of three measurements.

**Figure 4 molecules-29-04207-f004:**
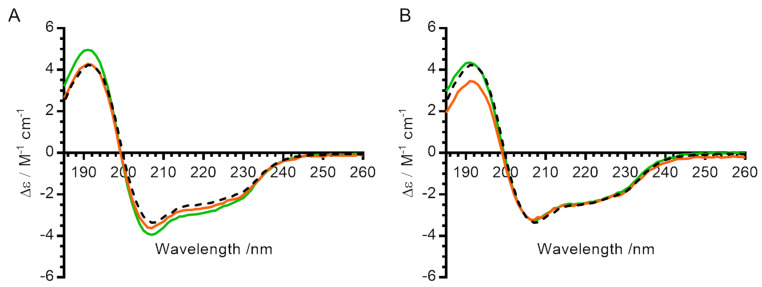
Effect of ensilication and heat treatment on secondary structure of lysozyme. (**A**) Circular dichroism spectra for lysozyme ensilicated in phosphate buffer. (**B**) Circular dichroism spectra for lysozyme ensilicated in PBS. All measurements were made in 10 mM phosphate pH 7. Black dashed line—control (untreated) lysozyme. Green—lysozyme after ensilication and desilication alone. Orange—lysozyme after ensilication, heat treatment for 1 h at 100 °C in oven, and desilication.

**Figure 5 molecules-29-04207-f005:**
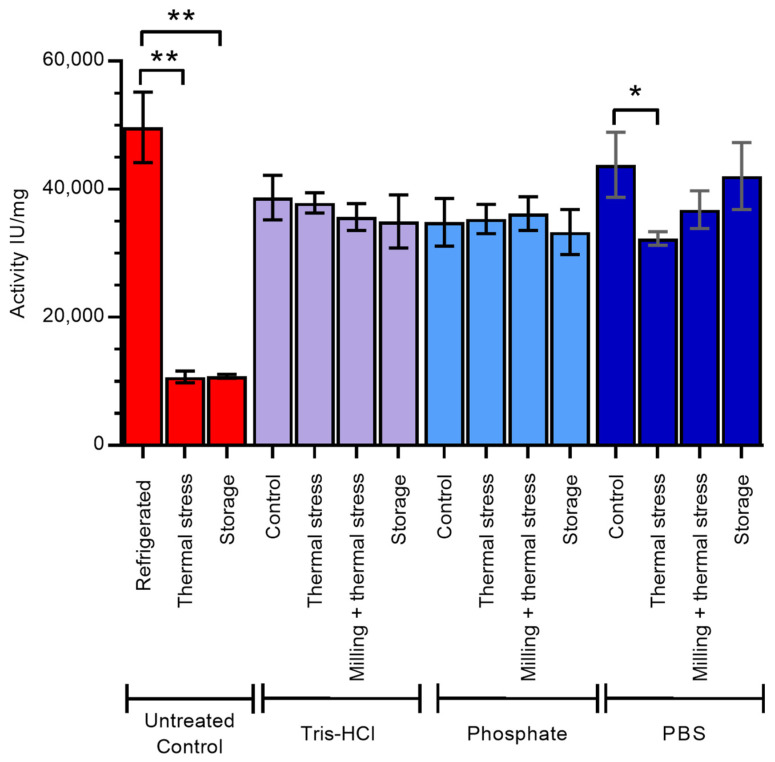
Enzyme activity of lysozyme after desilication from ensilication. The effect of long-term storage (18 months) at room temperature, thermal stress (1 h in oven at 100 °C), or milling and thermal stress combined while ensilicated, compared with control (ensilicated lysozyme dried overnight at room temperature), was determined. Untreated control samples are dry lyophilised lysozyme powder treated as described. Red—untreated control, purple—samples ensilicated in Tris-HCl, light blue—samples ensilicated in phosphate buffer, dark blue—samples ensilicated in PBS. Error bars indicate standard deviation of three measurements. * indicates significant difference (*p* < 0.05) in mean values and ** indicates significant difference (*p* < 0.01) in mean values, as determined by Tukey test.

**Table 1 molecules-29-04207-t001:** Physical and chemical comparison of ensilication buffers.

	Ensilication Buffer		
	50 mM Tris-HCl	50 mM Phosphate	PBS
Buffer pH	7.2	7.2	7.4
Buffer ionic strength /mM	45	118	166
Positively charged electrolytes (concentration /mM)	H^+^, (HOCH_2_)_3_CNH_3_^+^ (45)	H^+^, Na^+^ (84)	H^+^, Na^+^, K^+^ (158)
Negatively charged electrolytes (concentration /mM)	Cl^−^ (45)	H_2_PO_4_^−^, HPO_4_^2−^ (84)	H_2_PO_4_^−^, HPO_4_^2−^, Cl^−^ (158)
[Cl^−^] /mM	45	0	140
Particle diameter /nm *^a^*	420 ± 100	250 ± 30	460 ± 90

*^a^*: Errors indicate standard deviation.

## Data Availability

The original contributions presented in the study are included in the article. Numerical raw data and higher resolution images are deposited in Zenodo (DOI 10.5281/zenodo.13629114).

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
