# Peer review of "Thermal Stabilisation of Lysozyme through Ensilication"

_molecules, 2024, doi:10.3390/molecules29174207_

Round 1

Reviewer 1 Report

Comments and Suggestions for Authors

The authors of this research article aim to show that lysozyme mostly maintains its structure and activity of after process of ensilication and desilication. This research utilizes several complementary techniques, FTIR, CD, FE-SEM and UV-VIS, that allow the authors to reach a sounding conclusion. Furthermore, the authors clearly show that the chloride ion affect the size of the silica particles and therefore the decreases the loading efficiency.

This work is clearly written and well structured, it explains the importance of therapeutic transport and the process of the ensilication/desilication. Furthermore, it identifies the gap and complements the knowledge on these processes. A great aspect of this work is the usage of the complementary techniques that allow to reach a solid conclusion. This article mainly responds to the question that the ensilication buffer and the presence of chloride ions affect the ensilication process. Moreover, the authors show that the ensilitcation/desilication processes are viable and could decrease the energy usage during transport.

The presented data, in the form of figures, is clean and clearly presented and the captions are well written meaning the reader can reach the same conclusion as the authors.

It would be interesting to test Tris buffer without the chloride ions (e.g. Tris-SO4 or Tris-acetate) to determine if Tris affects the size of the silica particles. Other buffers, pH, and the presence of other anionic species or even additives that are commonly used in the preparation of therapeutics (e.g. amino acids, poly-ols and sugars) could also be tested.

I pose one comment and one question for the authors regarding the methodology.

1)      From line 239 to 242, the authors describe the procedures using “parts” for HCl, TEOS and lysozyme. I believe this is rather confusing and units of concentration should be used when working with stock solutions so procedures could be reproduced, otherwise one can assume that it will work at any given stock concentration.

2)      On line 243 the authors explain the desilication procedure and the working buffer. However, the work from reference 7 (“Thermal stability, storage and release of proteins with tailored fit in silica”), also written by two of the authors, states “Treatment using either fluoride or acid separately did not dissolve the silica or release proteins”. The submitted article states that the desilication procedure was performed using “25 mM Tris-HCl pH 7.2, 95 mM NaF” which is incompatible with the published literature. Is the desilication protocol presented in this work incomplete or it is a novel protocol?

On line 3, I believe the author name "Kumanov" should be read "Koumanov"

On line 234 the equation 3 is incomplete.

This article fits the scope of the special issue “Recent Advances in Lysozyme” in Medicinal Chemistry of “Molecules Journal” and I would accept this article with minor revisions.

Reviewer 2 Report

Comments and Suggestions for Authors

This work explored the effect of three biological buffers on the ensilication, storage, and desilication of lysozyme, finding that the ensilication buffer had minimal impact on lysozyme activity. The overall quality of the paper is great; however, a few small issues should be fixed.

  1. Line 68: Clarify what is meant by "drug loading" as there is no introduction on drugs in the introduction.

  2. Lines 66-71: This section includes results, which should not be in the introduction. The introduction should provide a brief literature review, indicate the research problem, state the research hypothesis, and verify the purpose of the study. Please amend this section to clearly state the hypothesis, problem, and research objective.

  3. The introduction needs more references, particularly to justify the novelty of the present approach.

  4. The Amide I band of the FTIR should be zoomed in to allow for a conclusion regarding the maintained protein secondary structure.

  5. Figure 2: Include a sample of silica alone for comparison.

  6. Report the particle concentration used in each test.

  7. Line 328: Indicate the name and source of the substrate used.

Round 2

Reviewer 2 Report

Comments and Suggestions for Authors

The authors have addressed all my comments. I agree with its current form for the publication.